# Contributions of cell behavior to geometric order in embryonic cartilage

**Sonja Mathias**[1]*, **Igor Adameyko**[2,3‡], **Andreas Hellander**[1‡], **Jochen Kursawe**[4‡]

**1** Department of Information Technology, Division of Scientific Computing, Uppsala University, Uppsala, Sweden, **2** Department of Physiology and Pharmacology, Karolinska Institutet, Solna, Sweden, **3** Department of Neuroimmunology, Center for Brain Research, Medical University of Vienna, Vienna, Austria, **4** School of Mathematics and Statistics, University of St Andrews, St Andrews, United Kingdom

‡ These authors contributed equally to this work as co-senior authors.
* sonja.mathias@it.uu.se

## Abstract

During early development, cartilage provides shape and stability to the embryo while serving as a precursor for the skeleton. Correct formation of embryonic cartilage is hence essential for healthy development. In vertebrate cranial cartilage, it has been observed that a flat and laterally extended macroscopic geometry is linked to regular microscopic structure consisting of tightly packed, short, transversal clonar columns. However, it remains an ongoing challenge to identify how individual cells coordinate to successfully shape the tissue, and more precisely which mechanical interactions and cell behaviors contribute to the generation and maintenance of this columnar cartilage geometry during embryogenesis. Here, we apply a three-dimensional cell-based computational model to investigate mechanical principles contributing to column formation. The model accounts for clonal expansion, anisotropic proliferation and the geometrical arrangement of progenitor cells in space. We confirm that oriented cell divisions and repulsive mechanical interactions between cells are key drivers of column formation. In addition, the model suggests that column formation benefits from the spatial gaps created by the extracellular matrix in the initial configuration, and that column maintenance is facilitated by sequential proliferative phases. Our model thus correctly predicts the dependence of local order on division orientation and tissue thickness. The present study presents the first cell-based simulations of cell mechanics during cranial cartilage formation and we anticipate that it will be useful in future studies on the formation and growth of other cartilage geometries.

## Author summary

In embryos, the initial skeleton is made out of cartilage. As the embryo grows, this cartilage needs to increase in size while correctly maintaining shape. A recent study revealed that for cartilage found in growing skulls, a flat sheet-like geometry is reflected in a distinct arrangement of cells at the microscopic level. Cells sharing a common ancestor are arranged into short columns such that the sheet grows in thickness by lengthening columns, and expands length-wise by adding new columns from single precursor cells. In

**Data Availability Statement:** All model specific code including scripts for data analysis and figure generation is available on GitHub at https://github.com/somathias/CartilageCBM/. All data generated by the numerical experiments have been uploaded

to figshare at https://doi.org/10.6084/m9.figshare.21731804.

**Funding:** SM and AH acknowledge funding from the NIH under grant no. NIH/2R01EB014877-04A1 and from the eSSENCE strategic initiatives on eScience. SM acknowledges a travel grant from the Anna-Maria Lundins stipend at Smålands Nation in Uppsala (grant nr. AMh2021-0081) to visit the University of St Andrews. IA was supported by ERC Synergy Grant 856529, Knut and Alice Wallenberg Foundation, Swedish Research Council, Austrian Science Fund, Paradifference Foundation, EMBO Young Investigator Program and Göran Gustafsson Foundation. The funders had no role in study design, data collection and analysis, decision to publish, or preparation of the manuscript.

**Competing interests:** The authors have declared that no competing interests exist.

this work we investigate the mechanical principles underlying column formation and insertion using a computational model that individually represents cells and their behavior. We confirm that arrangement of clonal columns perpendicular to the main expansion direction of the sheet requires oriented cell division. Moreover, we find that column order benefits from an increased amount of extracellular matrix between cells. Similarly, our model suggests that new clonal columns are able to insert themselves into pre-existing cartilage if sufficient matrix is available. Our model constitutes an important step to study cartilage formation and growth in different geometries which will be useful for understanding skeletal developmental disorders as well as for applications in tissue engineering.

## Introduction

Correct formation of embryonic cartilage is essential for healthy development. Cartilage provides structural support to the growing embryo and it serves as precursor to bone formation. A recent study investigating the cellular structure of cartilage in mouse embryos revealed that embryonic cartilage is highly geometrically ordered [1].

More specifically, cells inside growth plates of the cartilaginous skull are arranged in columns. These columns are oriented transversally and arranged adjacent to each other in the lateral direction, see Fig 1A for a schematic. For example, chondrocyte clones in the olfactory capsule of E17.5 mouse embryos form clearly visible columns (Fig 1B). The columns emerge from an initial, disordered, mesenchymal condensation which can be observed as early as E12.5. During the time span from E12.5 to E17.5, the finer features of the nasal part of the cranium are added through subsequent waves of new mesenchymal condensations, attaching to already existing cartilage. In general, once a thin, ordered sheet with a column length of 4–6 chondrocytes is formed, the sheet thickens as columns increase in length, as demonstrated in the embryonic olfactory capsule and inner ear of mouse embryos [1] (Fig 1C). However, the thickness increase in the embryonic cartilage (Fig 1D) is less pronounced than the simultaneous lateral expansion of the snout of the mouse (Fig 1E). This lateral growth relies on the insertion of new columns into the pre-existing cartilage [1]. It starts from the perichondrial stem-like cell layers on the tissue boundaries such that cells in one column are clonal, i.e. they typically are daughters of one perichondrial cell (Fig 1F and 1B). Ordered spatial cell arrangement was also reported in other embryonic cartilage structures, such as rod-like structures that precede the formation of long bones [1].

What mechanisms contribute to this high level of geometric order? Kaucka *et al.* showed that column formation relies on oriented cell division, which can be perturbed by ectopic activation of ACVR1, a receptor of Bone Morphogenetic Protein (BMP) [1]. These initial findings raise further questions. Are oriented cell divisions sufficient to transition from the disordered, mesenchymal condensation to tissues consisting of cell columns? What roles do cell behavior and mechanical interactions play in the formation and maintenance of columns? Is iterative growth in form of lengthening columns and insertion of new columns beneficial to the maintenance of the cellular order in the tissue, or does the tissue need to overcome mechanical challenges to enable this iterative growth?

Here, we aim to illuminate key principles governing embryonic cartilage formation and growth by simulating the process using a cell-based computational model. Cell-based computational models have a long history of helping to identify simple rules that contribute to the emergence of order and pattern formation in biological tissues. For example, placement of hair bristles in *Drosophila* can be explained by simple interactions between Notch and Delta

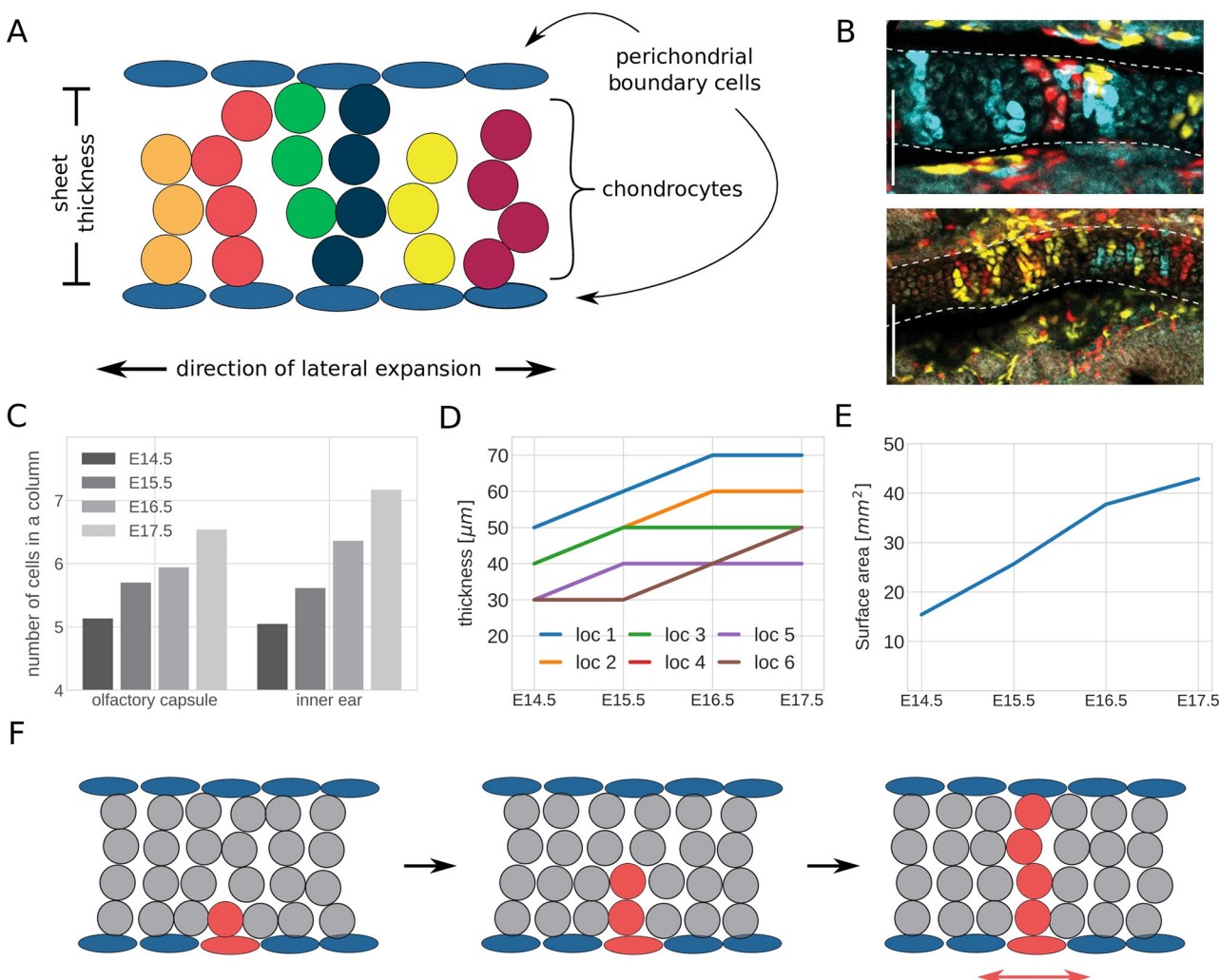

**Fig 1. Clonar column growth in sheet-like embryonic cartilage.** (A) Schematic of clonar column arrangement; (B) Chondrocyte clones at E17.5 in the olfactory capsule (up) and the inner ear (down) obtained via genetic tracing of neural crest cells at E8.5 using the Plp1-CreERT2/R26Confetti mouse lineage; The scale bars in the lower left corners equal 100 *μm*. (C–E): Measurements of cartilage geometry in four different developmental stages (E14.5-E17.5) (C) Number of cells per column in the olfactory capsule and inner ear; (D) Cartilage thickness measured at 6 different locations of the nasal capsule; (E) Cartilage surface area of the nasal capsule.; (F) Schematic of lateral expansion through clonar column insertion into pre-existing cartilage; Panels (B–F) are adapted from Figs 3, 3S2, 4 and 10 in [1], made available under a Creative Commons CC0 public domain dedication (https://creativecommons.org/publicdomain/zero/1.0/).

signalling [2, 3]. Similarly, mammalian blastocyst formation has been shown to rely on few simple principles, such as differential adhesion and signalling [4]. Cell shapes in growing epithelial tissues can be explained using simplified descriptions of cell mechanical properties and cell-cell interactions [5, 6]. Cell-based computational models have also been widely applied in cancer [7], angiogenesis [8], and other contexts.

Multiple types of cell-based computational models exist, and different modelling paradigms consider cells and their shape at varying levels of detail. For example, cell-centre based models consider cells as overlapping spheres [9], whereas vertex models represent epithelial cells as polygons [10]. These models both differ from on-lattice approaches in which space is represented by a lattice. Lattice sites can be occupied by individual cells in so-called Cellular Automaton models [11], or cells can extend across multiple lattice sites in Cellular Potts models [12].

Several mathematical and computational frameworks have been developed which provide modular and adaptable implementations for cell-based computational models. Examples of such frameworks are Chaste [13], CompuCell3D [14], PhysiCell [15] and Morpheus [16].

Here we use a cell-centre based model. Chondrocytes have round shapes that lend themselves to the approximation by spheres, which is an inherent component of such models. We implement our model in Chaste, which provides sufficient functionality to enable our studies, such as existing implementations for centre-based models, and the modularity to adapt tissue geometry, cell-cycle progression, and cell-cell interactions.

Our approach extends previous efforts on designing mathematical and computational models of cartilage. Kaucka *et al.* used a cellular automaton model to identify the necessity of oriented cell divisions for the maintenance of columns [1]. Since cellular automaton models operate on a lattice they cannot represent effects from mechanical interactions, and hence this model was not able to answer the further questions outlined above. In another study, Lycke *et al.* designed an off-lattice computational model of cartilage cells in embryonic femurs of mice, and investigated the distribution of mechanical loads between chondrocytes and ECM [17]. While this study included a careful consideration of cell and tissue mechanics, it did not consider cell division or other cell behaviors. Further models exist to study the formation of mesenchymal condensations in vitro [18, 19]. On the scale of a single cell, models have been designed to describe the molecular pathways that determine cell differentiation events in chondrocytes [20, 21]. On the tissue scale, several studies designed continuous mathematical models of embryonic cartilage mechanics that do not account for cell-cell interactions, for example studies on embryonic joint formation [22, 23] or endochondral ossification [24]. Similar continuum models are concerned with the maintenance and mechanics of adult cartilage [25–29]. Other models study cartilage generation in engineered tissues [30]. Despite these extensive efforts to accurately model cartilage in embryos and adults, a comprehensive computational investigation of column formation in growth plates remains missing.

Here, our simple, cell-based representation of cartilage formation reveals multiple insights. Similar to previous experimental findings [1], our model requires oriented cell divisions to establish columnar order in embryonic cartilage. We find that initial column formation benefits from space between progenitor cells in the lateral direction, and similarly from distance of these progenitor cells to the perichondrial boundaries. Our simulations correctly capture the fact that longer columns are more disordered, and that this disorder can be reduced if tissue thickness is increased incrementally. We find that oriented cell division and repulsive interactions between cells are not sufficient in our simulations to enable the intercalation of new cells into the tissue. In this case, our simulations suggest that the generation of space via the deposition of extracellular matrix (ECM) may be a further necessary ingredient.

The remainder of our paper is structured as follows. In the following 'Materials and Methods' section, we present our mathematical model and numerical methods. Then, we provide details and results of our computational investigations. Finally, we discuss implications of our results and avenues for future research.

## Materials and methods

In this section we describe the computational model of cartilage formation from initial aggregates of mesenchymal stem cells. Our model considers the effects of cell division, placement of progenitor cells, domain boundaries, and mechanical cell-cell interactions. Specifically, we use a cell-centre based model. In cell-centre based models, individual cells are represented as spheres [9, 31] that interact through mechanical forces and which are allowed to overlap. Our use of the cell-centre based model is motivated by the fact that the main cell type represented

in our model, chondrocytes, exhibit a characteristically rounded morphology [1, 32] (Fig 1B). In the following, we provide a mathematical description of the dynamics in the model, the force laws that we apply between cells, our chosen initial conditions, and our representation of the cell cycle model. Additionally, we introduce the *envelope projection area* as a measure to assess to what extent ordered, clonal columns are present in the model. Finally, we provide details of our numerical implementation and describe how to access the simulation source code.

## Cell motion is determined by interaction forces

We denote by $N(t)$ the number of cells that are present at time $t$. Cells are assumed to experience drag, but not inertia due to the small Reynolds number of cellular environments [33]. The velocity of the midpoint $\mathbf{x}_i$ of cell $i$ is therefore determined by the forces acting on it from its neighbours

$$\eta \frac{d\mathbf{x}_i}{dt} = \sum_{j \neq i} \mathbf{F}_{ij}. \tag{1}$$

Here, $\eta$ denotes the friction coefficient between the cells and the extracellular matrix as the surrounding medium. The sum includes all cells in the system, except cell $i$. The pairwise force $\mathbf{F}_{ij}$ is given by

$$\mathbf{F}_{ij} = F(\|\mathbf{r}_{ij}\|) \frac{\mathbf{r}_{ij}}{\|\mathbf{r}_{ij}\|}, \tag{2}$$

i.e. it points in the direction of the line $\mathbf{r}_{ij} = \mathbf{x}_j - \mathbf{x}_i$ connecting the centres of cells $i$ and $j$ and its sign and magnitude only depend on the distance between the cells.

We model the magnitude of pairwise interactions using a repulsion-only, piecewise quadratic force [15],

$$F(r) = \begin{cases} -\mu \left(1 - \dfrac{r}{s}\right)^2 & \text{if } r \leq s, \\ 0 & \text{otherwise.} \end{cases} \tag{3}$$

Here, $r = \|\mathbf{r}_{ij}\|$ denotes the distance between the cell pair and $\mu$ the spring stiffness, i.e. the strength of the repulsive interactions. Furthermore, cells that are further apart than the rest length $s$ do not exert forces on each other. Throughout, $s$ is defined as 1.0 cell diameter (see Table 1) and can be thought of as the sum of the radii of two interacting cells. If cells are within one rest-length of each other, they push each other away.

Our choice of this repulsion-only force, which does not include terms representing cell-cell adhesion, is informed by biological properties of the modelled chondrocytes. While cell adhesion plays a role in the formation of the initial mesenchymal condensation, cell adhesion molecules are not present once differentiation into cartilage starts [34]. Instead, the proliferating chondrocytes are surrounded by extracellular matrix preventing them from forming adhesive bonds with their neighboring cells [32], motivating our use of a repulsion-only force.

We do not expect the exact shape of the force function 3 to affect our results. This is based on previous findings in [35] and will be discussed further in the Discussion section.

## Cell proliferation

Our simulations account for cell division events, which we implement as follows. At each division event, the mother cell is removed from the simulation. Two daughter cells of equal radius

**Table 1. Model and numerical parameters along with their default values.**

| Parameter | Description | Value |
|---|---|---|
| $T$ | Simulation end time | 80 *a.u.* |
| $\Delta t$ | Simulation time step size | 0.0083333 *a.u.* |
| $\eta$ | Friction coefficient | 1.0 |
| $n_x$ | Number of cells in $x$-direction | 8 |
| $n_y$ | Number of cells in $y$-direction | 12 |
| $n_z$ | Number of cells in $z$-direction | 1 |
| $p_{max}$ | Maximum perturbation of initial coordinates | 0.1 $d$ |
| $u$ | $z$-coordinate of upper boundary plane | 3.5 $d$ |
| $l$ | $z$-coordinate of lower boundary plane | 0 $d$ |
| $n_{max}$ | Maximum number of cells per clonal envelope | 4 |
| $s$ | Rest length | 1.0 $d$ |
| $\mu$ | (Repulsive) spring stiffness | 20.0 |
| $g_1$ | Fixed duration of the first cell cycle phase | 3 *a.u.* |
| $g_2$ | Mean of the exponentially distributed duration of the second cell cycle phase | 10 *a.u.* |
| $r_0$ | Initial separation distance between daughter cells | 0.3 $d$ |
| $c$ | Scaling of distances between neighboring midpoints in the lateral $x$-$y$-plane | 1.075 |

The default values are used across all experiments unless specified differently for individual experiments. Length scales are measured in cell diameters $d$ which is set to $d = 1.0$ in all simulations. Time scales are measured in arbitrary time units *a.u.*.

are placed with a fixed separation distance $r_0$ between them such that the former position of the mother cell lies in the middle. The division orientation, i.e. the line along which the daughter cells are separated from the mother cell, is situation dependent. It can be chosen as a random direction (uniformly distributed in any direction) (Fig 2A), or along the fixed direction of the $z$-axis (Fig 2B), i.e. vertical (see discussion of geometry and initial conditions below). In this latter case, the $x$ and $y$ coordinates of the daughter cell are identical to those of the former mother cell. Throughout the paper, we will use vertically oriented cell divisions as in Fig 2B, except for two cases: the first computational experiment in the 'Results' section where we specifically study the effect of a random division orientation as in Fig 2A, and the second

**A** Random cell division direction **B** Oriented cell division along z-axis

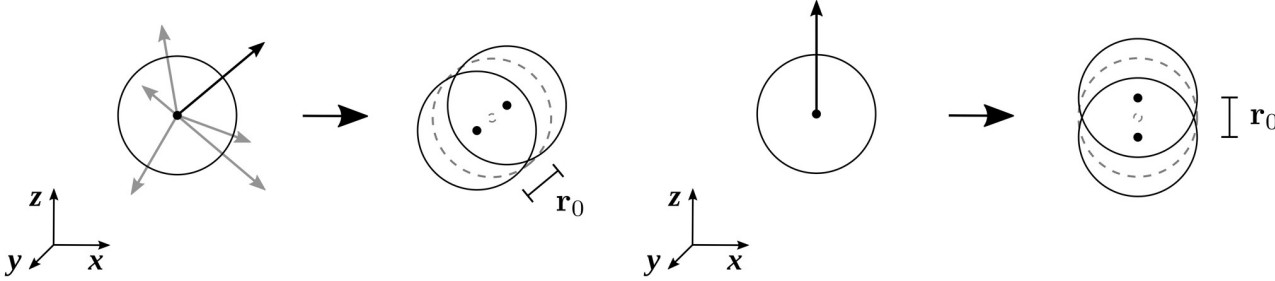

**Fig 2. Schematic of cell division implementation used.** When a cell is ready to divide, either a random cell division direction is drawn (A), or a cell division direction oriented along the $z$-axis is used (B). The two daughter cells are placed on a line going through the centroid of the former mother cell, and pointing in the drawn division direction. The cells are placed equidistant from the previous mother cell and separated from each other by the distance $r_0$.

computational experiment where we study the effect of perturbations added to vertical division orientations (details explained in place). We choose the value of $r_0$ to be 0.3 cell diameters.

These cell division events alter the system of ordinary differential Eq 1 by introducing new cells and changing neighborhood arrangements. In our simulations, cell proliferation is largely responsible for the dynamical behavior of the cell population in the following sense. The insertion of daughter cells leads to overlapping cell neighbors and thus results in localized tensions which will propagate through the cell population, pushing cells apart until the population reaches mechanical equilibrium.

To determine the timing of cell division events, we employ a cell cycle model containing two phases. The first cell cycle phase has a fixed duration $g_1 = 3.0a.u.$, while the duration of the second cell cycle phase is exponentially distributed with mean $g_2 = 10a.u.$ This choice of cell cycle model prevents synchronicity of cell divisions and thus ensures that neighboring cells do not divide simultaneously. This helps to maintain the quasistatic limit of our simulations (see discussion below). Similarly, including a fixed duration phase prevents immediate successive divisions. The volume of a cell does not change during the cell cycle.

## Definition of clonal envelopes

We call the collection of all cell offspring arising from one ancestor cell present in the initial configuration a *clonal envelope*. That ancestor cell may be a mesenchymal or perichondrial cell, see section 'Initial and boundary conditions' below. Kaucka *et al.* showed that the number of cells in each clonal envelope is related to chondrocyte maturation speed, as mature chondrocytes cease proliferation [1]. To reflect the existence of such a control mechanism (without modeling the biochemical details), we limit clonal envelopes to contain $n_{max}$ cells in our model, meaning that cells cease to proliferate once the size of their clonal envelope reaches this limit. Unless specified otherwise, we use $n_{max} = 4$.

## Initial and boundary conditions

To study both the initial formation and thickening of the cartilage sheet, and lateral growth through the subsequent insertion of new clonal envelopes into pre-existing cartilage, we use two different initial conditions which we describe in the following.

**Configuration (i): Cartilage formation from a mesenchymal condensation.** Biologically, the cartilage sheets considered here are formed from an initial condensation of mesenchymal cells [36]. The formation of the mesenchymal condensation itself is a complex, highly regulated process involving cell recruitment, migration, and condensation [36, 37]. Once progenitor cells are in place, they differentiate into chondrocytes which mature over a period of time and then cease to proliferate. During the maturation process, immature chondrocytes undergo multiple divisions. The initial conditions of our simulations are designed to reflect the mesenchymal condensations at the stage where progenitor cells have condensed and chondrocyte differentiation begins.

The spatial domain of our simulations also reflects the geometry of the biological tissue (Fig 3). We place 96 cells to generate an initial condensation 8 cells wide in the *x*-direction, 12 cells long in the *y*-direction, and comprising a single cell in the *z* direction. These dimensions of the initial mesenchymal condensation have been chosen large enough to collect robust statistics on the 'order' of each generated column, while at the same time being low enough to ensure simulation times of a single run do not exceed a few minutes. This short computation time ensures that parameter sweeps are computationally feasible. In the direction of lateral expansion, i.e. within the *x-y* plane, the cells are arranged on a honeycomb lattice (Fig 3B), in which

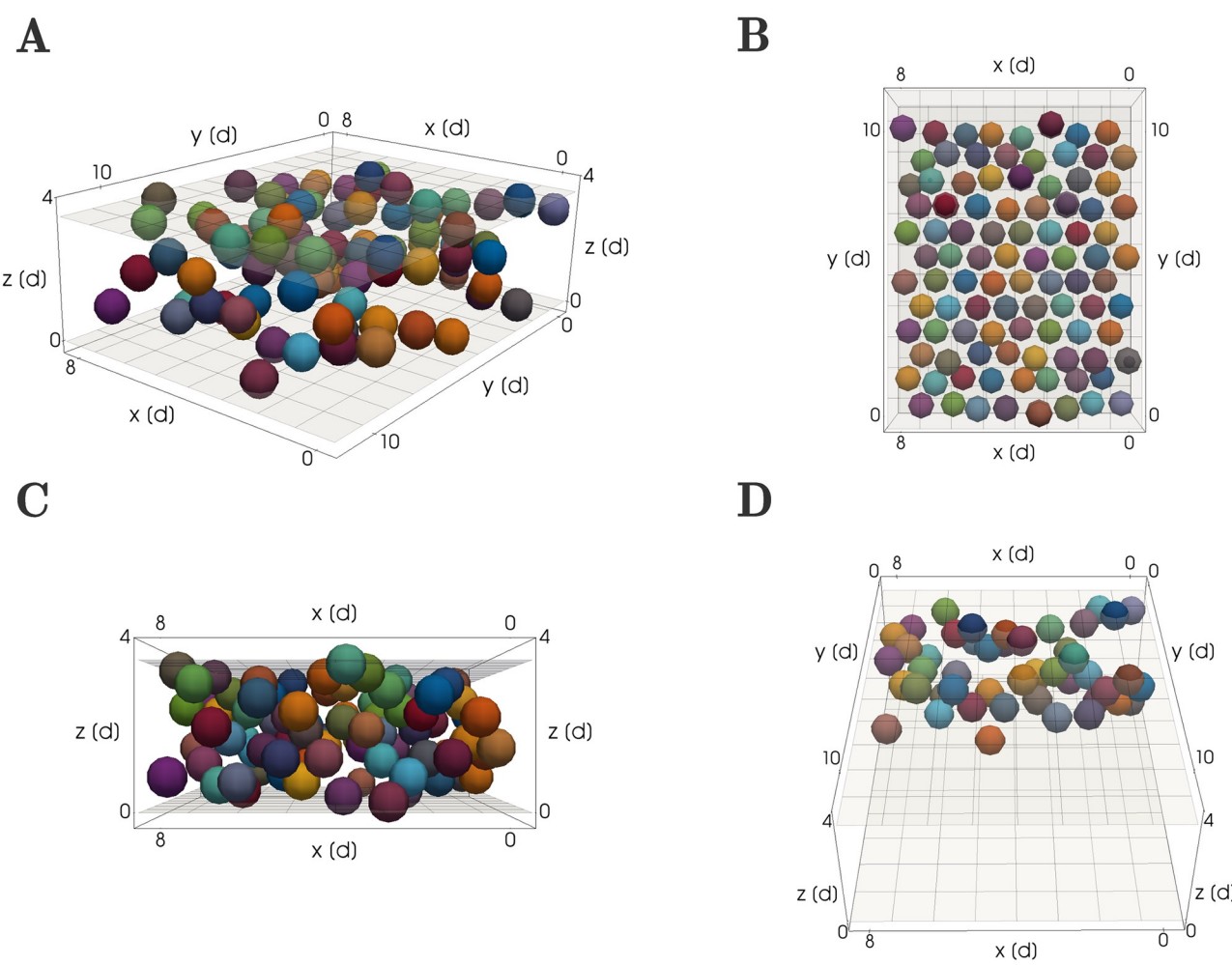

**Fig 3. Configuration (i): Cartilage formation from a mesenchymal condensation.** Cells were randomly colored and will pass on their color to their progeny. Cells located in close proximity have different colors so that different growing clonal envelopes will be easily distinguishable at later time points in the simulation. (A) General 3D view visualizing the overall shape of the mesenchymal condensation as well as the lower and upper boundary planes situated at their default values of $l = 0$ and $u = 3.5\ d$. (see Table 1) (B) View of the initial condition from above the cell arrangement in the lateral *x-y* plane. (C) Sideways view to visualize the spread of the mesenchymal cells in the transversal *z* direction, showing the positioning of cells between the upper and lower rigid boundary planes. (D) View of the cell population within a subsection of the simulation domain to show cell arrangements with in the tissue. Similar visualisations are used to investigate column formation throughout the Results section.

distances between neighboring cells are set to 1.075 *s*. Here, *s* denotes the rest length (see Eq 3), chosen by default as one cell diameter. This arrangement leads to cells being densely packed in the direction of lateral expansion, mirroring the biological requirement of a high cell density to initiate the differentiation into chondrocytes [32]. In the transversal direction *z*, we let individual cells be spread out between the upper and lower rigid boundary planes located at $z = l$ and $z = u$, as shown in Fig 3C. Specifically, *z*-locations are chosen uniformly at random in $z \in (l, u)$. These boundary planes model the mechanical influence of the surrounding perichondrial tissue. Note that we do not impose any boundary conditions in the *x-y* plane, i.e. cells are allowed to move freely in the *x* and *y* directions. Spreading of the cell population in these directions is naturally limited by the friction term in Eq 1, and by the absence of laterally oriented external forces.

Additionally, a perturbation drawn uniformly from $[0, p_{max}]$ is added to each coordinate of all cell midpoints to allow for biologically realistic variations on the cell positions. Note that the rigid boundaries of the condensation are enforced at the cell midpoints, meaning that parts of the cell's spheres in visualisations can cross the boundary planes, as visible in Fig 3. The value of $p_{max} = 0.1\,d$ is chosen small enough as to not significantly alter the geometrical arrangement of the initial mesenchymal condensation.

To visually evaluate columnar order inside the tissue, rather than on the tissue boundary, we consider a view looking along the $y$ axis onto the middle of the condensation, and ommitting cells from the visualisation that are not within the 'back' half of the tissue (Fig 3D). This enables us to observe the geometrical shape of clonal envelopes in the middle of the sheet. In the Results section we use this view when visualizing example snapshot configurations of the mesenchymal condensation and the forming cartilage at different time points.

**Configuration (ii): Cartilage growth through insertion of new clonal envelopes into a pre-existing sheet.**   In [1], the authors identified a clonal relationship between chondrocyte columns and individual perichondrial cells, suggesting that during cartilage expansion new clonal envelopes were seeded from the perichondrial cells located at the periphery of the pre-existing cartilage. Building on this hypothesis, we designed a second simulation setup to study the insertion of new columns as follows. We modeled a pre-existing sheet through layers of cells (Fig 4A). These were arranged as transversal columns according to an underlying hexagonal lattice in the $x$-$y$ direction (Fig 4B). The number of cells in the $x$ and $y$ dimension were the same as for the mesenchymal condensation. We used four layers of chondrocytes and added two layers of perichondrial cells, one above and one below the chondrocyte sheet, still within the rigid boundary planes (Fig 4C). Perichondrial cells differed from chondrocytes in their cell division dynamics and orientation, dividing parallel to the sheet asymmetrically into one perichondrial daughter cell and one chondrocyte daughter cell. A fraction of the perichondrial cells in both layers were "activated" i.e. chosen to divide once in order to insert new columns into the pre-exisiting sheet (colored in Fig 4A and 4B). Their progeny cells inherited the coloring to enable identification of clonal envelopes. Perturbation of the initial coordinates, initial spacing between cells in the $x$-$y$-plane and the properties of the rigid boundary planes above and below the sheet were chosen identical to initial configuration (i).

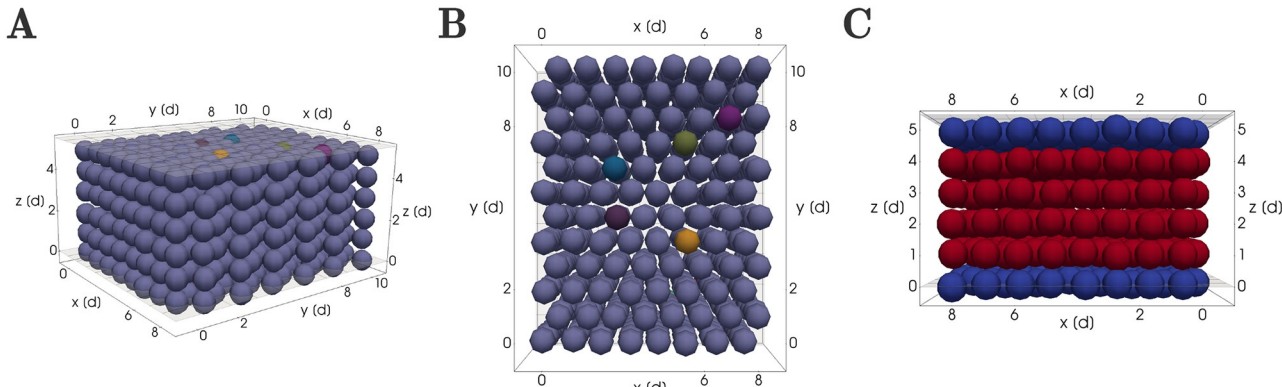

**Fig 4. Configuration (ii): Cartilage growth through insertion of new clonal envelopes into a pre-existing sheet.** (A) Three dimensional view. (B) View from above of the cell arrangement in the lateral $x$-$y$ plane according to a perturbed honeycomb mesh. The differently colored perichondrial cells in the top layer seen in (A) and (B) divide giving rise to clonal envelopes. (C) Sideways view to visualize the layers of chondrocytes (in red) and perichondrial cells (in blue) between the upper and lower rigid boundary planes.

## Model parameter values and quasistatic limit

In this section we briefly introduce and motivate our parameter choices concerning the dynamics of our simulations. Specifically, we discuss the values of the repulsive force strength $\mu$ and the friction coefficient $\eta$ in Eqs 1 and 3, as well as the simulation duration $T$.

We choose a repulsive force strength of $\mu = 20.0$ and a friction strength of $\eta = 1.0$. The ratio between these two parameters $\eta$ and $\mu$ determines the mechanical relaxation time scale in the simulations, since both parameters only occur as multiplicative constants in Eqs 1 and 3. These values are chosen such that the system operates within a quasistatic limit, i.e. such that the tissue can achieve a stable configuration between consecutive cell divisions.

The choice of simulating a quasistatic limit is motivated by biological timescales. Cell cycle duration of proliferating chondrocytes in the clonal envelopes is on the order of 24 hours as estimated from Fig 4 in [1]. We expect mechanical relaxation to occur on a time scale of minutes, i.e. roughly three orders of magnitude faster than cell division.

To reflect the fact that our time scales are only loosely based on real physical units, we measure time in *arbitrary time units* which we abbreviate as *a.u.*. Simulation end time is chosen as 80 *a.u.*, which is sufficiently long for final cell population configurations to reach a mechanical equilibrium. Mean cell cycle duration is chosen as 13 *a.u.* to ensure that simulation wall time remains sufficiently short while conserving the quasistatic equilibrium. Here, the duration is split between the phases as $g_1 = 3$ *a.u.* and $g_2 = 10$ *a.u.* (see description of the cell cycle model above).

All parameter values used in the simulation of our model are summarized in Table 1, along with their default values.

## Randomness in the model

Randomness enters the model in four ways, firstly through the perturbation of the initial cell coordinates, secondly through the distribution of the G2 cell cycle phase (and as a result the distribution of cell division times), thirdly, if cell division is not set to take place in an oriented fashion, through random cell division directions, and lastly, through the choice of activated perichondrial cells in initial configuration (ii). We therefore run numerical experiments for each parameter setting with 8 different random seeds and average over the results.

## Metric for evaluating order in the cartilage sheet

To evaluate the quality of the clonal column growth we measured the shape of each clonal envelope. For each envelope we calculated its projection area by multiplying the maximum deviation in both $x$ and $y$ direction. We then averaged over all envelopes to obtain the average envelope projection area $a$ as

$$a = \frac{1}{\#\text{envelopes}} \sum_{\mathcal{E}:\ \text{clonal envelope}} \left| \max_{\text{cells}\in\mathcal{E}} x - \min_{\text{cells}\in\mathcal{E}} x \right| * \left| \max_{\text{cells}\in\mathcal{E}} y - \min_{\text{cells}\in\mathcal{E}} y \right|. \tag{4}$$

The smaller $a$ is, the more column-like the geometrical shape of the average clonal envelope is.

## Simulation procedure

Simulation of our cartilage growth model proceeds as follows. The cell population is created according to the initial configuration and cells are initialized at a given position in space (see section 'Initial configuration' above). Time is then advanced in discrete time steps until the end time is reached. At each time point, both cell states and positions are updated in a two step process. First, all cells advance through the cell cycle by the elapsed time. When a cell is ready

to divide, it is removed from the simulation and replaced by two new daughter cells. The duration of each cell's G2 phase is drawn individually from an exponential distribution at cell initialization (see section 'Cell proliferation' above).

Once all possible cell division events have been carried out, the positions of all cells in the population are updated. Eq 1 is solved numerically using the forward Euler method [38]. Cell midpoint coordinates are updated by adding the sum of the current force interactions scaled with the current time step length $\Delta t$ as

$$\mathbf{x}_i^{\text{new}} = \mathbf{x}_i^{\text{old}} + \Delta t \sum_j \mathbf{F}_{ij}^{\text{old}}. \tag{4}$$

If the new position $\mathbf{x}_i^{\text{new}}$ is beyond one of the rigid boundaries of the domain, it is projected back onto the boundary by changing its $z$-component to the $z$-position of the boundary. We use the time step $\Delta t = 0.0083333$ *a.u.*, the default value in the Chaste software [13]. Once cell positions are updated, simulation time is advanced and the next time step begins. This loop continues until the simulation end time is reached.

## Results

We apply our cell-based computational model to understand how cell behaviours and mechanical interactions contribute to the creation and maintenance of cell columns inside embryonic cartilaginous growth plates. In this section we investigate how the distinct geometrical shape of clonal columns can form from initially condensed mesenchymal ancestor cells, before moving on to ask how column intercalation into pre-existing cartilage may be achieved.

### A repulsion-only force and oriented cell division enable column formation

When sheet-like cartilaginous elements are formed from an initial mesenchymal condensation e.g. in the nasal capsule of the embryonic mouse, clonal envelopes arrange themselves very robustly into highly ordered columnar shapes. The mesenchymal condensations giving rise to these sheet-like elements are very thin, spanning only 1 or 2 cell diameters in height [1]. Hence, progenitor cells arising from different ancestors are likely to be obstructed by neighbours in the lateral direction ($x$-$y$ in Fig 3), but not in the transversal direction ($z$ in Fig 3). This motivated us to ask whether a regular spacing of initial mesenchymal ancestor cells combined with randomly oriented divisions may be sufficient to induce the growth of clonal columns. To answer this question, we arranged cells according to the initial configuration (i) described in Fig 5A, see Section 'Initial and boundary conditions'. We then simulated cartilage formation as described in Section 'Simulation procedure'. We let cells divide in randomly chosen directions until each clonal envelope contained 4 cells. Rigid boundary planes above and below the condensation constrained cell positions to the developing sheet, as a substitute for the mechanical influence of the surrounding tissue.

We found that this simulation was unsuccessful in generating ordered columns. Clonal aggregates observed for this case in Fig 5B do not show a recognizable geometrical order, and clonal envelopes are overlapping. This is in agreement with previous findings by Kaucka *et al.* [1], who combined mathematical modeling and experimental interventions to show that oriented cell divisions, guided by a gradient of bone morphogenetic proteins (BMP), are essential for correct column formation *in vivo*. Hence, we concluded that our first simulation in Fig 5B does not result in column formation since we neglected this important biological component.

To address this lack of biological realism in our simulations, we proceeded to include oriented cell divisions. As the exact molecular mechanism of the gradient formation remains unknown, we simulated oriented cell divisions by deterministically setting the division

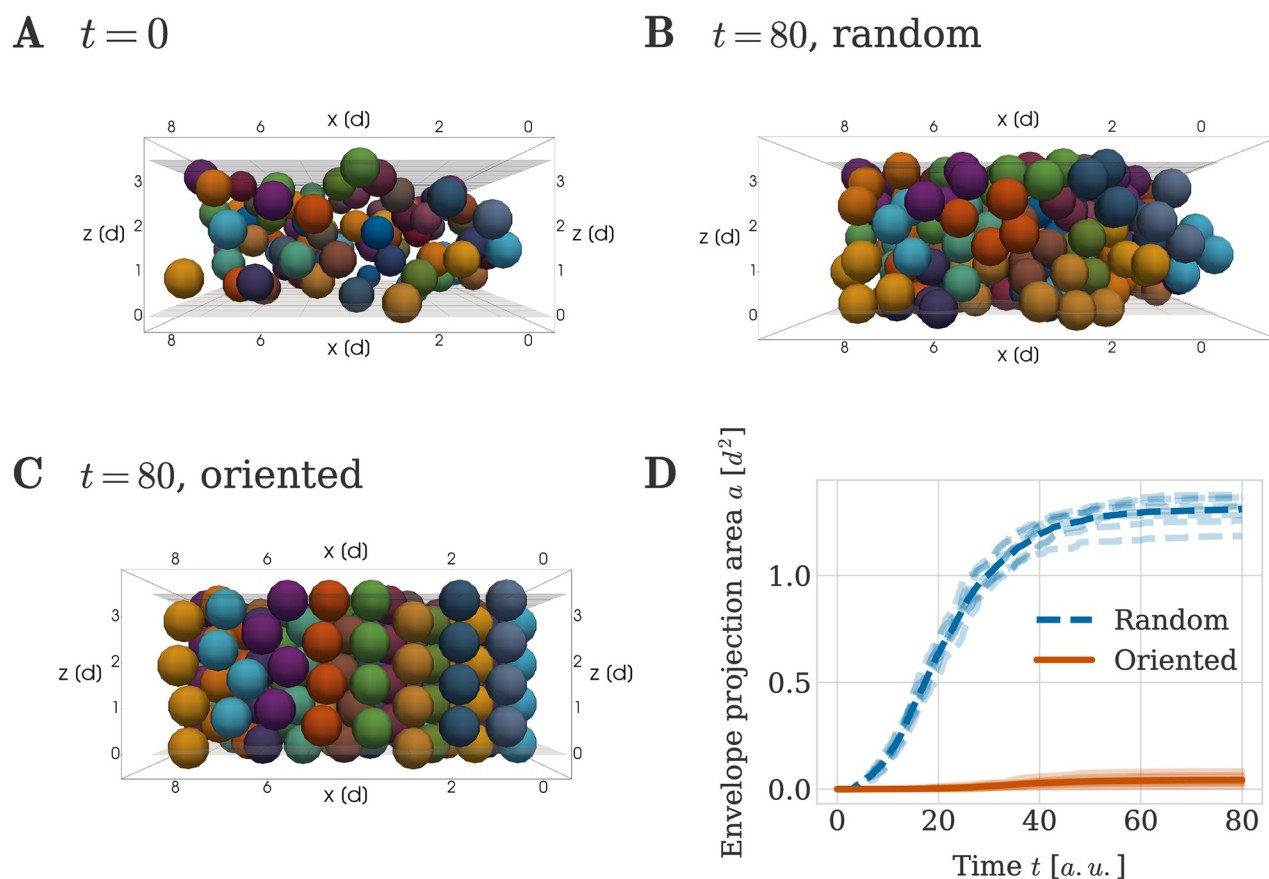

**Fig 5. Impact of oriented and random cell division directions on column formation.** (A) Initial configuration of mesenchymal condensation (view cuts through the middle), used for both experiments with oriented and random cell division directions. (B, C) Example configurations at $t = 80$ $a.u.$ when using (B) random and (C) oriented cell division directions. Same view as in (A). Colors are consistent across plots (A), (B) and (C) with cells within a clonal envelope inheriting their color from their ancestor. Ancestor cells located in close proximity were assigned different colors so that clonal envelopes were easily distinguishable. (D) Envelope projection area over time for oriented and random cell division directions. The thick line denotes the average over 8 independent repeats of the simulation. The individual data of each simulation are plotted with decreased opacity.

direction in our computational model to align with the $z$-axis, so that all divisions are oriented in the transversal direction of the tissue. As a result, oriented cell division happened transversely to the main expansion direction of the sheet. As predicted, inclusion of this mechanism led to clonal envelopes forming column-like structures (Fig 5C).

The visually observed differences in panels (a) and (b) can be quantified using the envelope projection area metric introduced in the 'Materials and Methods' section for measuring the order of cartilage columns. In brief, the metric averages over all clonal envelopes at a given timepoint in the simulation, and it is designed to reflect the lateral spread of individual clonal envelopes in the $x$-$y$ plane. If envelopes on average have a column-like shape, the metric is small, and conversely, unordered shapes result in larger values. Fig 5D depicts the envelope projection area as a function of time for both the case with randomly chosen cell division directions (Fig 5B) and the case when cells divided in an oriented fashion (Fig 5C), for 8 simulations in each case. The envelope projection area increased rapidly for the random cell division orientation, whereas it stayed small for transversally oriented divisions. Specifically, at the end of the simulation the envelope projection area under random division orientations was 30 times larger than the envelope projection area under oriented cell divisions. It was hence clear

that across multiple, independent simulations, clonal envelopes consistently grew in column-like structures when allowed to divide vertically along the *z*-direction, in contrast to them exhibiting no geometrical order when division directions were chosen randomly. We conclude that oriented cell division in combination with a repulsion-only force (no adhesive forces necessary) is sufficient for column formation from a mesenchymal condensation.

Our findings agree well with previously reported findings from experimental investigations. Kaucka *et al.* reported that cell divisions are oriented by a gradient of Bone Morphogenetic Protein (BMP) [1]. Perturbing how cells 'read' the local BMP gradient misoriented cell division. This lead to clonal envelopes that were disordered and not columnar. In practice, this experiment was based on constitutive overactivation of ACVR1 in individual cells, a receptor for BMP that is located on the cells surface. The fact that oriented cell divisions are necessary for column formation *in vivo* and *in silico* confirms that our model assumptions capture essential properties of chondrocyte dynamics.

Having observed simulated envelope projection areas in Fig 5, we now asked what values this measure may take in experimentally observed columnar structures. To do so, we estimated the envelope projection area from published figures in [1]. First, we manually extracted coordinates of cell centres in flourescently labelled clonal envelopes from figures published in [1] using Fiji [39]. In this way, we identified the cell centres of cells in 12 different clonal envelopes from olfactory, pharyngeal, and inner ear cartilage. From these data, we estimated envelope projection areas of clonal envelopes containing four or five cells to be $0.23 \pm 0.2\ d^2$ (see S1 Appendix for a detailed description). This value was well below the simulated envelope projection area for randomly oriented cell divisions of $1.31 \pm 0.06\ d^2$ and slightly larger than the simulated envelope projection area for vertically oriented cell divisions of $0.04 \pm 0.02\ d^2$ (both measured at $T = 80\ a.u.$). This highlights that columns are ordered in the experimentally reported images. Since our experimentally measured values of the envelope projection area were larger than simulated values for oriented cell division, we proceeded to ask what other factors may influence column order.

## Column order is sensitive to perturbations in division orientation

When considering oriented cell divisions in Fig 5, cells in our simulations divided strictly along the vertical direction, i.e. parallel to the *z*-axis. This is a simplification of chondrocyte behaviour, as cell division directions can be expected to stochastically vary. Findings by Kaucka *et al.* indicated that this is indeed the case [1]. The authors measured the relative position of daughter cells using EdU incorporation experiments. They found that, while daughter cell pairs had a tendency to align vertically, their alignment differed from the vertical axis by a standard deviation of up to 15˚ ([1], Fig 4, panels D-F and K-L). These observations motivated us to investigate how small variations in division orientation can affect column order.

We defined division orientation between two daughter cells using spherical coordinates (Fig 6A). Points on the unit sphere are determined by the azimuthal angle $\varphi \in [0, 2\pi]$ and the polar angle $\theta \in [0, \pi]$. In this framework, strictly vertically oriented cell division directions correspond to a polar angle of $\theta = 0°$ for any angle $\varphi$. To investigate the influence of small deviations from the vertical division direction onto column order, we allowed $\theta$ to randomly vary at each division event, with a maximum value of $\theta_{max}$. This was achieved by drawing a value for $\cos(\theta)$ from a uniform distribution between $\cos(\theta_{max})$ and 1, and calculating $\theta$ from the sampled value. The azimuthal angle $\varphi$ was drawn from a uniform distribution between 0 and $2\pi$. Drawing $\cos \theta$ rather than $\theta$ from a uniform distribution ensured that the endpoints of division vectors were evenly spaced on the unit sphere [40].

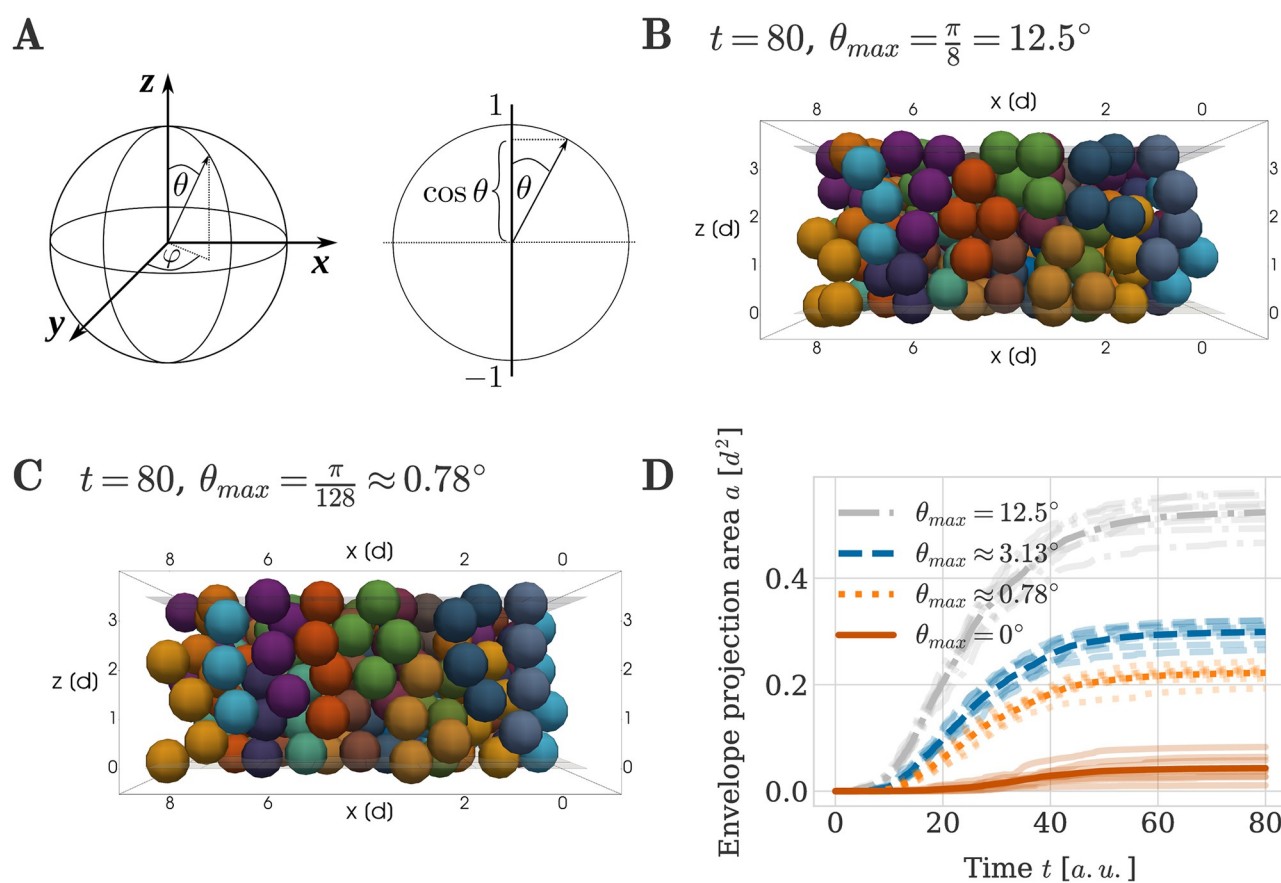

**Fig 6. Impact of perturbations in division orientation on column formation.** (A) Left: In spherical coordinates a point on the unit sphere is defined by the azimuthal angle $\varphi$ and a polar angle $\theta$. Right: Cosine of the polar angle $\theta$. Random values of $\theta$ were generated by drawing $\cos\theta$ from a uniform distribution between [$\cos\theta_{max}$, 1], ensuring evenly spaced samples on the unit sphere with $\theta < \theta_{max}$ [40]. (B, C) Example simulation snapshots at $T = 80$ *a.u.* for two different values of the maximum polar angle $\theta_{max}$. (D) Envelope projection area over time for different values of the maximum division angle $\theta_{max}$. The thick line denotes the average over 8 independent repeats of the simulation. The individual data of each simulation are plotted with decreased opacity.

We analysed how different magnitudes of variation in the division angle influence column order by running simulations with varying $\theta_{max}$. The largest perturbation angle we considered, $\theta_{max} = 12.5°$ ($= \pi/8$), was informed by Kaucka *et al.,* who found that alignment angles of clonal cell doublets have a standard deviation between 10° and 15° [1]. Simulations with this value led to a visual decrease in order of the clonal envelopes (Fig 6B, compare with Fig 5C), yet clonal envelopes were more column-like than in the case of fully random orientation (compare with Fig 5B). Moreover, we found that even small perturbations in the division orientation can have a large effect on column order. For example for $\theta_{max} = 0.78°$ ($= \pi/128$) in Fig 6C, the resulting clonal envelopes were still visibly less straight than in the case of strictly vertical division orientations (compare with Fig 5C).

Quantifying these observations using the envelope projection area (Fig 6D), we observe that using $\theta_{max} = \frac{\pi}{8} = 12.5°$ resulted in an envelope projection area of $0.52 \pm 0.03$ $d^2$ at time $T = 80$ *a.u.*, which is larger than the value of $0.23 \pm 0.2$ $d^2$ measured on experimental data. In the simulations considered here, $0.23$ $d^2$ corresponds to a smaller maximum angle, closer to the data for $\theta_{max} \approx 3.13°$ ($0.30 \pm 0.02$ $d^2$) and $\theta_{max} \approx 0.78°$ ($0.22 \pm 0.01$ $d^2$). This latter value is considerably larger than the envelope projection area we observe when

$\theta_{max} = 0$ ($0.04 \pm 0.02$ $d^2$), further highlighting that even small perturbations in the division direction can lead to a noticeable difference in column order.

Note, that the angle distribution measured by Kaucka *et al.* is measured on cell doublets at unknown time intervals after division occurred [1]. Thus, these distributions measure a compound effect of mitotic spindle orientation and cell displacement after division. Hence, the experimentally measured angle distributions cannot be used directly to define a specific $\theta_{max}$ in our simulations.

## Column growth benefits from more extracellular matrix between cells

Having identified how cell behaviours can affect the growth of column-like structures from initial configurations, we next asked how robust this mechanism is to changes in our chosen initial condition. We hypothesised that cartilage formation is not only governed by dynamic cell behaviours but also by the cell and tissue configuration at the start of the process.

To this end, we investigated the impact of the amount of extracellular matrix dictating the spacing between cells in the initial spatial configuration of the population. We considered the scaling of the cell arrangement in the lateral *x-y* plane as depicted in Fig 7A. Biologically, increasing the distance between the individual ancestor cells with a scaling parameter $c > 1.0$ can be interpreted as there being more extracellular matrix between the mesenchymal ancestor cells. Note, that Fig 7A does not depict random perturbations that are added to each cell centre location (see section 'Initial and boundary conditions'). We included these perturbations to avoid numerical artifacts that one may expect from perfectly symmetric initial conditions.

Again, we let cells divide in a perfectly oriented fashion until clonal envelopes contained four member cells, while leaving all others simulation aspects unchanged to the previous simulation. For a scaling parameter value of $c = 1.0$, corresponding to a minimal amount of extracellular matrix, clonal envelopes exhibited column-like structures, yet the order of the column was decreased and columns were less straight (compare Fig 7B to Fig 5C). Increasing the scaling parameter to values above our default value of $c = 1.075$, e.g. to a value of $c = 1.1$, increased the smoothness and straightness of the columns formed by the clonal aggregates (Fig 7C). Again, this visual finding was quantified by the envelope projection area metric $a$ (Fig 7D). Taken together, these experiments demonstrate that the robustness of ordered column formation increases with larger initial spacing between cells.

Our interpretation of this effect is that increased spacing between columns minimises mechanical interactions between neighbouring columns, thus promoting column-straightness. This reduction in mechanical force as $c$ is increased can also be observed from the force law 3: the maximal force that two cells may experience between each other in the initial configuration when $c = 1.1$ is four times smaller than the maximal force when $c = 1.0$. This can be seen by considering that the largest value of our random perturbation in the initial condition is $p_{max} = 0.1$, and hence the closest possible distance between two initial cells when $c = 1.0$ is 0.8, whereas this distance would be 0.9 for $c = 1.1$. Inserting these values into 3 leads to a factor four difference.

## Column growth benefits from distance to the perichondrial boundary

We next proceeded to analyse the influence of the initial distance to the perichondrial boundary on the geometrical order of the formed clonal envelopes. In our simulations, the perichondrial boundary is represented by rigid planes at the positions $z = u$ and $z = l$, see section 'Initial and boundary conditions'. In addition to the initial lateral spacing between cells, the distance to these boundaries is a second biologically relevant aspect of our initial condition.

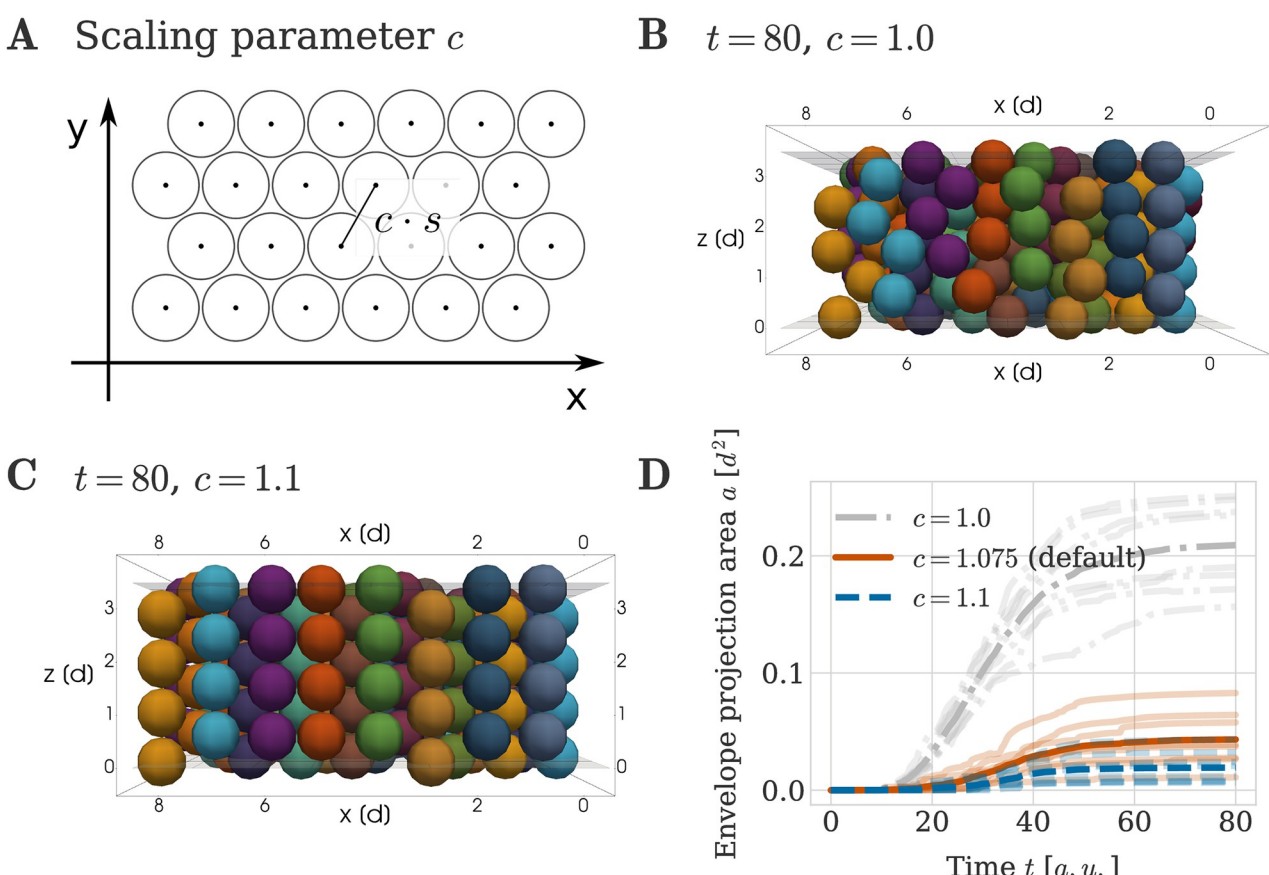

**Fig 7. Impact of amount of extracellular matrix on column formation.** (A) Visualization of scaling parameter $c$ determining the initial spacing between cells in the $x$-$y$ plane. The simulated mesenchymal condensation used in our simulations has 8 cells in $x$- and 12 cells in $y$-direction. Not depicted are random perturbations of maximal magnitude $p_{max} = 0.1$ $d$ that are added to the depicted cell centres (see section 'Initial and boundary conditions') (B, C) Example simulation snapshots at $T = 80$ $a.u.$, visualized as a crosssection through the centre of the mesenchymal condensation. (B) A scaling of $c = 1.0$ was used. (C) A scaling of $c = 1.1$ was used. (D) Envelope projection area over time for different scaling values $c$. The thick line denotes the average over 8 independent repeats of the simulation. The individual data of each simulation are plotted with decreased opacity.

The perichondrial boundary is a layer of cells surrounding the chondrochranium in sheet-like cartilages [1]. This layer is only a few cells wide and formed from mesenchymal ancestors located at the boundary of the mesenchymal condensation (see Fig 1A). For the process of endochondral ossification in the context of long bone development, it is well understood that the perichondrium plays an important role through several functions such as providing signalling cues controlling proliferation and differentiation of the chondrocytes within the condensation [41, 42], giving rise to cells that establish the bone collar as well as promoting the formation of blood vessels in the bone [43]. The perichondrium is structurally different to the cartilage as the perichondrial cells exhibit a flat morphology and are situated in a matrix characterized by a horizontal arrangement of collagen fibers [44]. We here model the mechanical influence of the surrounding tissue—including the perichondrial cell layers—on the developing cartilage through the use of rigid boundary planes. Having cells located away from the boundary can be interpreted as there being an increased amount of soft hyaline matrix.

To understand the influence of the matrix between cells and the perichondrial boundary on column formation we generate alternative initial conditions where, instead of spreading the cells of the mesenchymal condensation evenly between the upper and the lower boundary

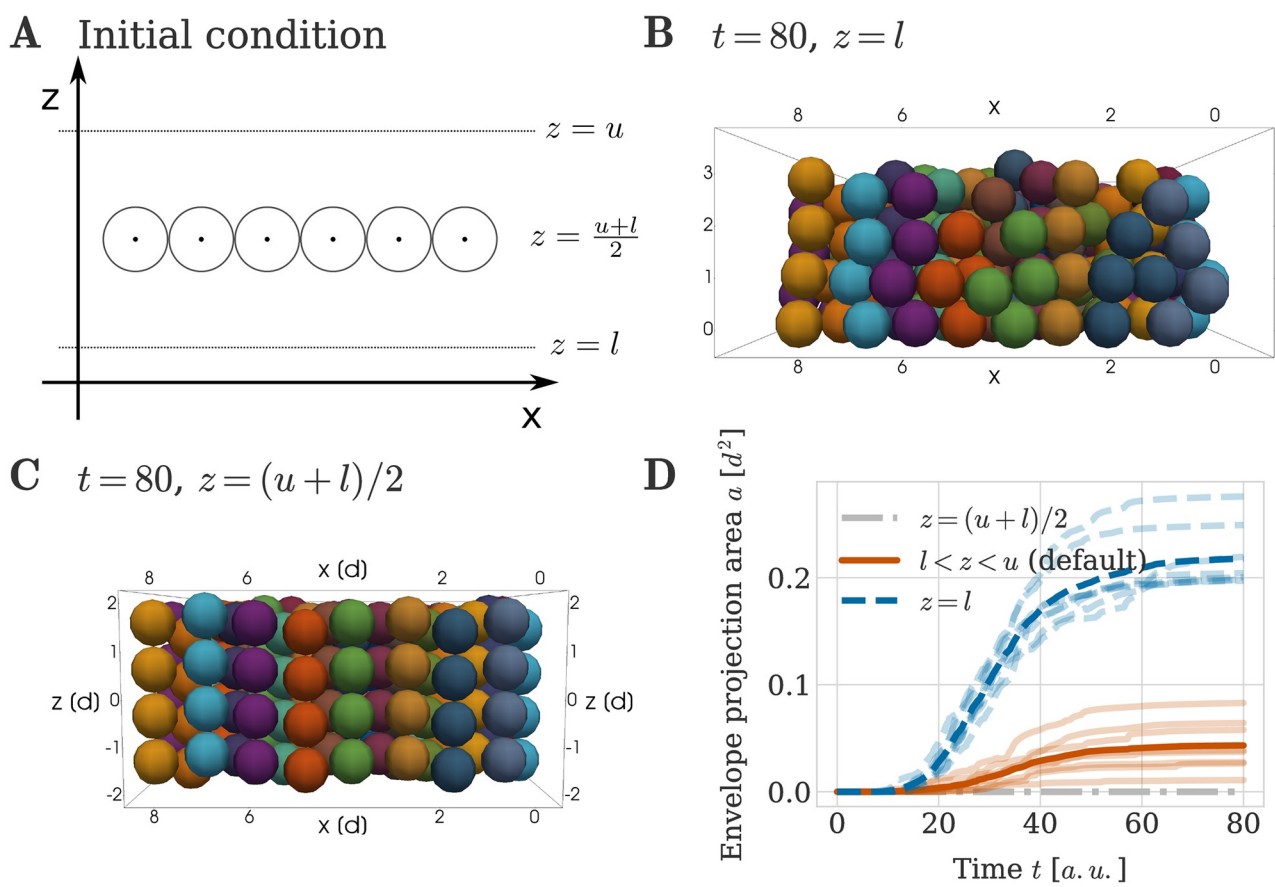

**Fig 8. Impact of distance to perichondrial boundary on column growth.** (A) Visualization of initial configuration. (B, C) Example simulation snapshots at $T = 80$ *a.u.* for all cells of the mesenchymal condensation being placed at (B) the lower perichondrial boundary ($z = l$) and (C) the middle between upper and lower boundary ($z = (u + l)/2$). (D) Envelope projection area over time for the different initial configurations. Semi-transparent lines denote the results from individual repeats of the simulation and the thick line denotes the average over all repeats.

planes, we let all mesenchymal ancestor cells be situated in a single layer (Fig 8A). In a first simulation, we placed cells at the lower perichondrial boundary $z = l$ and then let them divide in an oriented fashion. This resulted in clonal envelopes with visibly less ordered geometrical shapes than in previous simulations where the initial $z$–position was randomly chosen (Figs 8B and 5C). In a second simulation, we arranged all mesenchymal cells in a flat sheet situated at the middle between the boundary planes, i.e. at maximum distance from both boundaries, $z = (u + l)/2$ and obtained nearly perfectly straight column growth. This finding was confirmed by quantification through the envelope project area in Fig 8D, which is largest (i.e. columns are most disordered) when cells are initialised close to the perichondrial boundary. We conclude ECM and mesenchymal space between cells initiating column formation benefits column straightness.

## Trade-off between column length and order

Cartilage sheets are found in different parts of the developing skull such as in the nasal capsule, basisphenoid and the inner ear. Depending on their location their thickness differs with basisphenoid and inner ear cartilage being on average thicker than olfactory cartilage (see [1], Fig

8, panels B, D, F and H, results for the littermate wild type control). This motivated us to study how the geometrical order of the clonal shapes depended on the thickness of the sheet.

Biologically and in our simulations, the thickness of the sheet is controlled by the number of clones per envelope. We hence varied the maximum number of cells $n_{max}$ allowed in each envelope and adjusted the height $u$ of the upper boundary plane in the initial condition accordingly to ensure sufficient space for fully ordered columns (Fig 9A). We simulated column growth starting from our default mesenchymal configuration in which $z$-coordinates are drawn uniformly from the distance between the (adjusted) boundary planes, and allowed individual chondrocyte divisions according to our cell cycle model until the size of their clonal envelope reached $n_{max}$. Visual inspection of example simulations for $n_{max} = 6$ (Fig 9B) and $n_{max} = 8$ (Fig 9C) showed that for thicker sheets the geometrical order of the clonal shape decreased. Columns were more likely to be two cells wide when consisting of 8 cells (Fig 9C), although we stress that clonal envelopes continued to show a clear orientation transversal to the main lateral direction of expansion. Quantifying the results through the use of the envelope projection area confirmed that column order decreased with sheet thickness (Fig 9D).

These experiments suggested a trade-off between cartilage thickness and order in the cellular micro-structure of the cartilage. Indeed, visual inspection of long columns in Kaucka

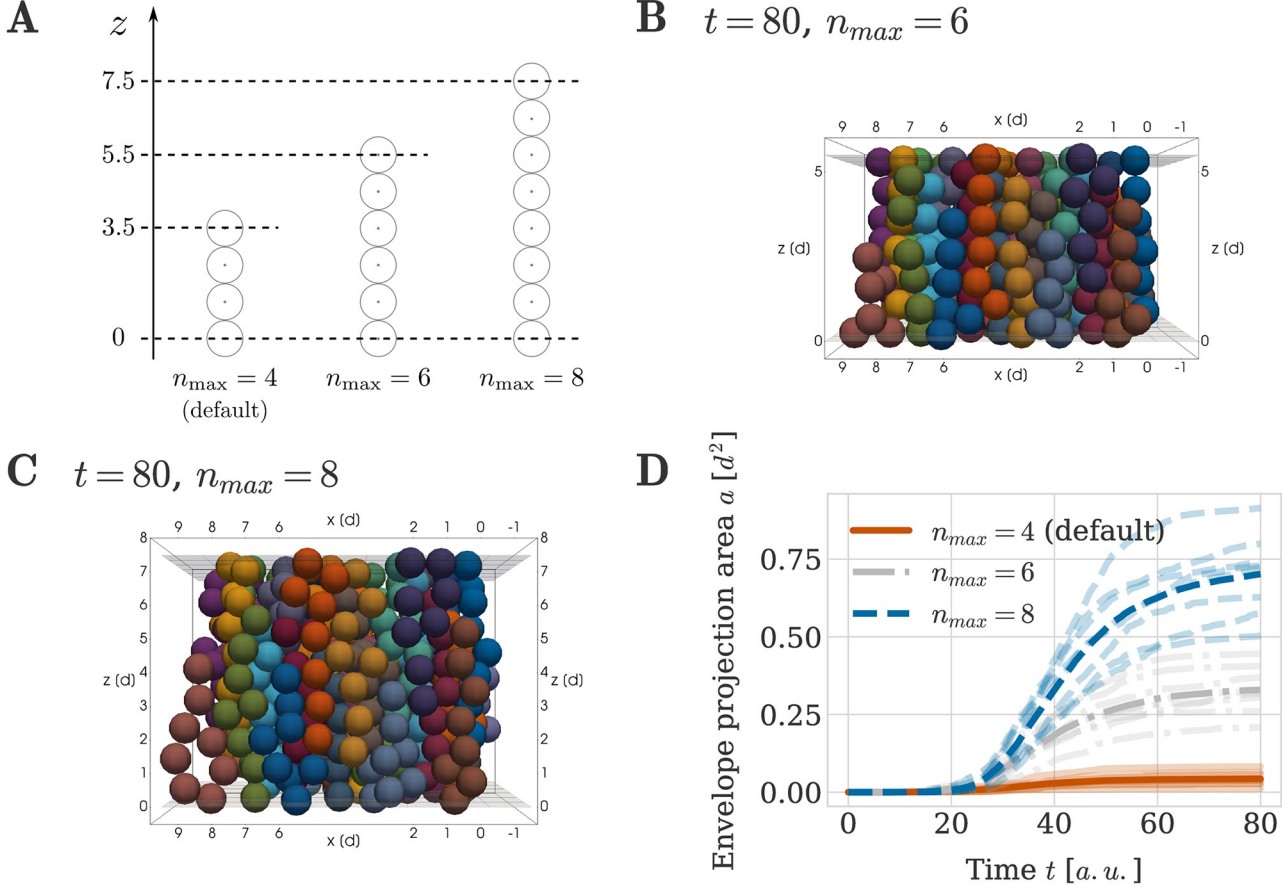

**Fig 9. Impact of sheet thickness on column order.** (A) Location of upper rigid boundary plane $u$ for different clonal envelope sizes $n_{max}$. (B, C) Example simulation snapshots at $T = 80$ *a.u.* for clonal envelope sizes being limited to (B) $n_{max} = 6$ cells and (C) $n_{max} = 8$ cells. (D) Envelope projection area over time for different clonal envelope sizes. Semi-transparent lines denote the results from individual repeats of the simulation and the thick lines denote the average over all repeats.

*et al.* illustrated that they are less ordered than shorter columns, (see e.g. [1], Fig 8, panels B, D and F).

## It is more efficient to keep order by increasing column length, than to grow larger columns from scratch

To accommodate the significant growth during development, cranial cartilage sheets in the mouse embryo need to be scaled accurately both in longitude and in thickness. As discussed in the previous subsection, growing thicker sheets through continuous proliferation may result in a decrease in geometrical order with increasing number of cells per clonal envelope. In plate-like cartilage sheets observed in [1], the growth of long columns was reported to be iterative, i.e. individual adjacent columns in one sheet grow to a fixed, uniform length, before additional cells are added to each column through cell division (see also Fig 1C–1E). This raised the question whether such iterative tissue thickening is beneficial to the order of individual columns.

To answer this question, we ran a simulation using a step wise time-dependent function for the limit of cells per envelope $n_{max}$ (Fig 10A, red line). Initially, the limit was chosen as 4 cells per envelope and, after a sufficiently long waiting time of $t_w = 50$ *a.u.*, this limit was then increased to 6 cells. This time point $t_w$ was chosen such that all clonal envelopes had time to grow to the limit of 4 cells and to mechanically relax before $t_w$ passed. As comparison, we used simulations in which the limit $n_{max}$ was chosen as 6 cells from the beginning (Fig 10A, blue line). Comparing example snapshots for the different proliferative profiles, we observed that if growth happened in sequential proliferative phases, envelopes had a higher chance of spanning the entire height between the lower and upper rigid boundaries (Fig 10b) at the end of the simulation. In contrast, if clones were continuously proliferating up to the final $n_{max}$ value, shapes often resulted in shorter columns two cells wide (Fig 9B). These qualitative differences were confirmed quantitatively with a smaller average envelope projection area in the case of iterative column growth (Fig 10C). Hence, our simulations indicated that iterative growth of columns observed in [1] by Kaucka *et al.* is beneficial for the maintenance of column order.

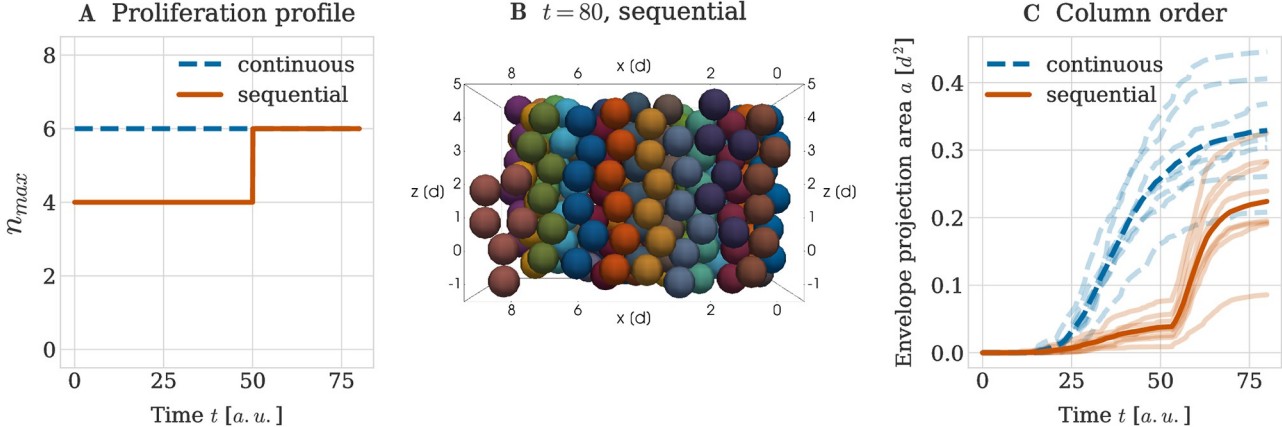

**Fig 10. Impact of sequential proliferative phases on column order.** (A) Profile of the limit on the number of cells per clonal envelope $n_{max}$ over time with either a single continuous proliferative phase (blue line) or two sequential proliferative phases with a step wise increase in the limit (red line). In either case the goal is to grow clonal columns 6 cells high. (B) Example simulation snapshot at $T = 80$ *a.u.* for the sequential proliferative profile. (C) Envelope projection area over time for the two different proliferative profiles (continuous and sequential). Opaque lines denote the results from individual repeats of the simulation and the thick line denotes the average over all repeats.

In addition to finer features of the developing mouse cranium being added by the induction of new mesenchymal condensations, the existing cartilage grows in size as well, while accurately keeping shape and proportions. In [1], the authors show that the complex structure of transversal columns in sheet-like cartilage allows for this to be done accurately by introducing new columns into the already existing sheet. We now apply our understanding about how columns form initially to study the principles of this secondary scaling process.

## Oriented cell division is not sufficient for column insertion

We hypothesised that the principles sufficient for initial column formation should also enable scaling of the sheet through column intercalation. To test this hypothesis, we simulated the insertion of new columns using initial configuration (ii) (see Section 'Initial and boundary conditions', Fig 4). These initial configurations were designed to represent a tissue in which column formation had already happened, and fully formed, well-ordered columns are initially present. In contrast to our previous simulations, these new initial configurations explicitly account for the existence of perichondrial cells at the perichondrial boundaries above and below the sheet. Nine perichondrial cells were then randomly activated across both perichondrial layers to initiate the insertion of new columns. Each of these activated perichondrial cells divided once with a division orientation parallel to the main expansion direction of the sheet, thus creating a new chondrocyte cell. The new cell then started dividing according to our cell cycle model with transversally oriented cell divisions, with chondrocyte cell daughters. Clonal expansion proceeded until the maximum number of cells per clonal envelope $n_{\mathrm{max}} = 5$ was reached, which was chosen so that potentially formed columns could span the the full height of the simulated sheet. The original perichondrial cell was not counted to this limit of $n_{\mathrm{max}}$ cells.

With this setup, clonal envelopes were able to insert themselves into the pre-existing sheet, but not in the shape of clearly visible columns (Fig 11A and 11B). Instead they consistently formed aggregates that were typically two cells wide and only reached to the middle of the sheet. Quantification via the envelope projection area metric confirmed that the geometrical order of the clonal envelope shape was decreased compared to a five cell column grown directly from the initial mesenchymal condensation (Fig 11C). This lead us to conclude that

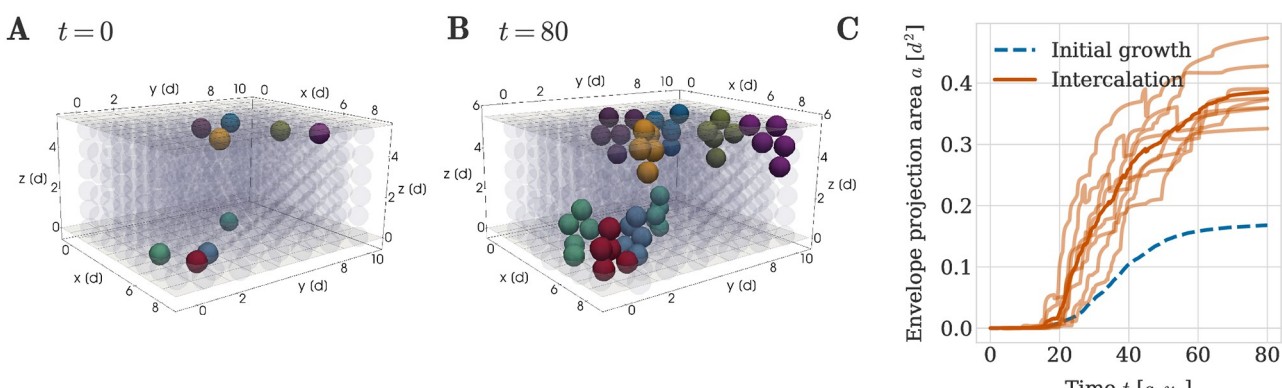

**Fig 11. Intercalation of clonal envelopes into pre-existing cartilage sheets.** An example simulation result is shown at (A) $t = 0$ and (B) $t = 80$ *a.u.*. Cells in the pre-existing cartilage are shown in grey with low opacity for better visibility of the shape of new clonal units. Different colours represent different clonal envelopes, such that cells belonging to the same clonal envelope share a colour. (C) Envelope projection area over time for the intercalation of clonal envelopes into a pre-existing sheet with no available space. Semi-transparent orange lines denote the results from different random seeds and the thick orange line denotes the average over all random seeds. For comparison the average projection area for clonal columns comprised of 5 cells grown directly from the initial mesenchymal condensation is shown in both panels in blue (denoted by 'initial growth').

the combination of orientated cell divisions and a repulsion-only force between cells is not sufficient to insert well-formed columns into existing cartilage.

### New columns can grow into existing cartilage if there is matrix

Our previous results indicated that column formation benefits from increased amounts of extracellular matrix, manifesting as gaps in the spatial arrangement of the mesenchymal ancestor cells in configuration (i). We hypothesized that the same principle holds true for the insertion of new clonal envelopes into pre-existing cartilage. To test this hypothesis, we considered a portion of the cartilage sheet described in configuration (ii), measuring 5 columns wide and 5 columns deep and deleted most central column (Fig 12A). We then chose a perichondrial cell next to the missing column to divide into the available space. Its chondrocyte progenity was then able to grow into a clearly columnar shape within the cartilage sheet (Fig 12B), resulting in a well-ordered column spanning the full height of the carilage sheet. We used the projection area metric on the inserted clonal envelope across multiple simulation repeats to confirm that, if there was space in the pre-existing cartilage, the average geometrical order of the clonal envelope was similar to a column grown directly from the initial mesenchymal condensation (Fig 12C).

To conclude, our results suggest that a possible mechanism for chondrocyte column intercalation involves existing chondrocytes secreting extracellular matrix before new columns are inserted into a cartilage sheet. This mechanism may be experimentally tested by flourescently labelling extracellular matrix components or otherwise measuring matrix secretion. Our simulations thus further imply an important role of extracellular matrix for the generation and maintenance of embryonic sheet-like cartilage.

## Discussion

How cells interact to coordinate morphogenesis is a key question in developmental biology. Here, we investigated this question at the example of embryonic cartilage in growth plates of the skull. Motivated by previous findings in mouse embryos, we studied how cells may arrange into columns from initially non-columnar mesenchymal condensations.

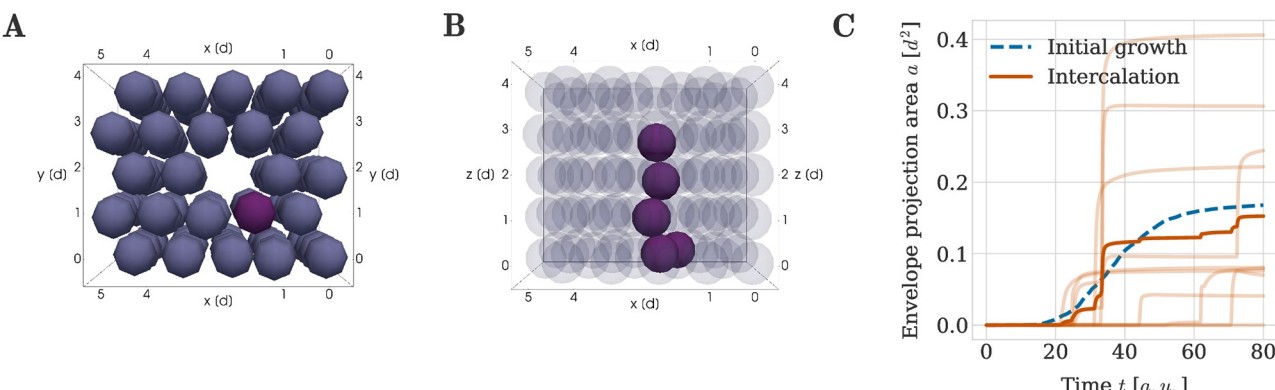

**Fig 12. Intercalation of clonal envelopes into pre-existing cartilage sheets with space made available beforehand via the secretion of extracellular matrix.** An example simulation result is shown at (A) $t = 0$ from below and (B) at $t = 80$ *a.u.* from the side. The purple perichondrial cell in (A) divided into the space provided and seeded the clonal envelope seen in (B). (C) Envelope projection area over time. Semi-transparent orange lines denote the results from individual repeats of the simulation and the thick orange line denotes the average over all repeats. For comparison the average projection area for clonal columns comprised of 5 cells grown directly from the initial mesenchymal condensation is shown in both panels in blue (denoted by 'initial growth').

Using a cell-centre based computational model, we confirmed that oriented cell divisions are necessary for column formation. We found that column formation benefits from space between progenitor cells in the mesenchymal condensation. Our model indicates that tradeoffs between column length and order can be mitigated by iterative growth. We identified that oriented cell divisions are insufficient to ensure that new columns can be inserted into existing tissues, and suggested that extracellular matrix may generate space before new columns are grown.

Our cartilage growth model complements a previous mathematical model of cartilage sheet formation presented in [1]. This previous mathematical model was a so-called Cellular Automaton model, in which the tissue was represented by a lattice, and each cell was allowed to occupy one lattice site. Unlike this previous model, the model developed here explicitly lets cells push on each other during mechanical relaxation, without relying on ad-hoc rules for lattice occupancy that are hard to relate directly to biological function and to parameterize. Using an off-lattice approach also eliminates possible grid artifacts and offers more flexibility in describing mechanical interactions. Importantly, the present model is also able to implicitly represent the presence of extracellular matrix by continuously varying the amount of space between cells. These benefits of the cell-centre based model enabled us to go beyond the previous modeling approach and investigate how mechanical interactions and extracellular matrix can contribute to robustness of ordered column formation as well as studying the principles for column intercalation.

The main simplifying modeling assumptions made by the cell-centre based model include (i) the spherical representation of cell shape, (ii) the assumption that cell interactions occur pairwise and (iii) the assumption that forces between cells can be approximated by our chosen repulsive interaction force. For our application of cartilage growth and formation, modeling cells as spheres in assumption (i) is motivated by the rounded morphology of chondrocytes as seen e.g. in Fig 1B. Similarly, as the chondrocytes are embedded in extracellular matrix, no cell-cell adhesion takes place between them. This can be seen for example in Fig 1B, where there are visible gaps between cells. Additionally, it has been reported in chick-embryos that N-Cadherin, a molecule responsible for cell-cell adhesion, is present in mesenchymal cells, but not mature chondrocytes [34]. These observations support the validity of pairwise mechanical interactions in assumption (ii) as well as our choice of a non-adherent repulsion-only force as part of assumption (iii). Regarding the shape of the force law in assumption (iii) we have chosen a piecewise quadratic force [15], see Eq 3. We do not expect this specific choice of force function to qualitatively affect our results. This is supported by previous findings in [35], which compared multiple force functions that are typically used in cell-centre based models, such as the piecewise quadratic force [15], the cubic force [45] or the generalized linear spring force [13]. Through this comparison, Mathias *et al.* found that all these different force functions can easily be parametrized such that their biological behavior at the population level agrees well in repulsion-dominated settings [35].

We note that our simulations do not include volume growth of cells during the cell cycle. This is a commonly made assumption in cell-centre simulation frameworks [13, 15]. See [35], section 2.5, for a further discussion of the topic.

In designing our initial condition, we have assumed that chondrocyte stem cells are initialised such that two initial columns would not intersect vertically. This is motivated by biological observations indicating that newly formed mesenchymal condensations are very thin and only contain one or two layers of cells (see [1], video 1). Our initial conditions further assume a noisy hexagonal packing of chondrocyte stem cells in the $x - y$ direction, which allowed us to simulate a scenario of high stem cell density. Yet, it will require further experimental investigation to confirm these assumptions and identify the range of cell

arrangements inside early mesenchymal condensations from which cartilage can be formed.

Our computational model further confirms previous findings from [1] that oriented cell division dynamics play an important role for enabling column growth. Our implementation of oriented cell division is a simplification of cell-dynamics in cartilage sheets, it may hence be improved in future versions of the model. Specifically, from the imaging data in [1], it is not distinguishable whether the division itself is really oriented, or if the initial cell division direction is random and then daughter cells slide around each other until the doublet is oriented transversely to the sheet. Biologically, a possible mechanism for this could be chemotaxis along the BMP gradient, since in other systems it has been shown that chemotaxis can be induced in chondrocytes using BMP [46]. This chemotaxis could be mediated through adhesion to the ECM, since chondrocytes are known to express integrins, which are molecules responsible for cell-matrix adhesion [47]. Hence, if the second case is true, oriented divisions in our model may be inaccurately implemented, since we deterministically set the division direction itself parallel to the $z$-axis, i.e. perpendicular to the main expansion direction of the sheet. An initial observation supporting the assumption that divisions are directly oriented in this way is that we do not observe un-oriented doublets in the imaging data presented in [1]. Hence, any sliding of daughter cells would need to take place quickly, which may lead to a tissue dynamics which are still well-represented by our implementation. In addition, in contrast to the model used in [1] we do not consider any noise in the cell division orientation. We believe that this additional detail is not required in our model, and it would introduce unnecessary model parameters. Specifically, doublets of daughter cells are already slightly misaligned in our simulations due to the influence from neighbouring cells. The source of this misalignment is noise that we have included in the initial arrangement of the progenitor cells in our initial configuration (see Section 'Configuration (i): Cartilage formation from a mesenchymal condensation'). The mechanical influence of slightly misaligned neighbors on daughter cells leads in practice to small perturbation in the orientation of each doublet. Thus our simulations naturally account for the influence of non-perfectly aligned daughter cells on column growth.

Our results on column insertion indicate that the deposition of ECM is a major component of cartilage morphogenesis. Specifically, additional space between cells that can be generated through the deposition of ECM may facilitate the maintenance of columns, as well the insertion of new columns into existing tissues during growth. This is consistent with the fact that chondrocytes are known to secrete large amounts of ECM [48] and express integrins, which are molecules responsible for cell-matrix adhesion [46]. Presumably, simulations such as those in Fig 12, in which space for a whole new column is available at once, are unrealstic. Instead, the mechanism for column insertion may involve deposition of the ECM at the 'tip' of a growing column.

Our model does not explicitly represent the extracellular matrix. Instead our model implicitly captures its effect through two mechanisms. Firstly, space between the cells in our configurations is directly related to the amount of matrix between them. Secondly, our mechanical governing Eq 1 account for mechanical interactions between cells and their microenvironment through the friction term on the left hand side of the equation. We believe that this overdamped force law realistically describes interactions between non-adhesive cells and the ECM. In future versions of the model, it would be straightforward to extend our model to study anisotropy in the properties of the extracellular matrix by varying the friction coefficient $\eta$ both in space or in time. As an example, the cartilage itself and the perichondrium surrounding it exhibit differences in the composition of their extracellular matrix and hence differ in their mechanical properties. The matrix surrounding the chondrocytes within the cartilage has a high amount of proteoglycan aggregates and collagen type II, making it more soft, whereas

the matrix around perichondrial cells is comprised of horizontally aligned collagen fibers of type I [44, 49]. An extension of our current model could then take the different mechanical properties of these two tissues into account by assigning different drag coefficients (with eta being smaller for the softer cartilage) to the different types of cells. Other, more explicit modeling of the extracellular matrix e.g. by introducing another type of (smaller) interacting particle would also be feasible within the framework of cell-centre based models.

Our model paves the way for future computational studies of cartilage development that may reflect longer durations of bone morphogenesis and thus allow us to simulate the full geometry of embryonic bones until larger structures such as growth plates of the skull are fully formed. In these models, dynamically changing, rather than fixed, boundary conditions may be desirable. Our methods may also enable studies of bone morphogenesis in different geometries. Kaucka *et al.* reported that cartilage contains transversely oriented columns not only in growth plates, which were discussed here, but also in rod-like cartilage that will later turn into digits or ribs [1]. In these tissues, circular transversal cross-sections of the cartilage contain columns that are slightly bent and have varying lengths. We believe that our presented computational framework for testing verbal hypotheses on cartilage formation may reveal insights into morphogenesis of such long cartilage structures with varying geometries, and help investigate how the corresponding bone structures may be formed robustly. Specifically, computational modelling may help answer the question of how growth is coordinated to achieve correct scaling in the transversal and lateral directions of growth.

The ability to accurately simulate the formation of cartilage and bone in varying geometries will be a crucial step towards advances in tissue engineering. Simulations of the type presented here may also provide insights into skeletal developmental disorders, such as achondroplasia [50].

## Supporting information

**S1 Appendix. Calculating envelope projection areas from published data.** In-depth description how the experimental envelope projection area used for comparison was calculated from figures depicting biological data published in [1].
(PDF)

## Author Contributions

**Conceptualization:** Sonja Mathias, Igor Adameyko, Andreas Hellander, Jochen Kursawe.

**Formal analysis:** Sonja Mathias.

**Funding acquisition:** Sonja Mathias, Andreas Hellander.

**Investigation:** Sonja Mathias.

**Methodology:** Sonja Mathias, Jochen Kursawe.

**Project administration:** Sonja Mathias, Andreas Hellander.

**Software:** Sonja Mathias.

**Supervision:** Igor Adameyko, Andreas Hellander, Jochen Kursawe.

**Validation:** Sonja Mathias.

**Visualization:** Sonja Mathias.

**Writing – original draft:** Sonja Mathias, Jochen Kursawe.

**Writing – review & editing:** Sonja Mathias, Igor Adameyko, Andreas Hellander, Jochen Kursawe.

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
