## [Decision Letter · Decision Letter 0]

28 Apr 2023

Dear Miss Mathias,

Thank you very much for submitting your manuscript "Contributions of cell behavior to geometric order in embryonic cartilage" for consideration at PLOS Computational Biology.

As with all papers reviewed by the journal, your manuscript was reviewed by members of the editorial board and by several independent reviewers. In light of the reviews (below this email), we would like to invite the resubmission of a significantly-revised version that takes into account the reviewers' comments.

We cannot make any decision about publication until we have seen the revised manuscript and your response to the reviewers' comments. Your revised manuscript is also likely to be sent to reviewers for further evaluation.

Sincerely,

Philip K Maini

Academic Editor

PLOS Computational Biology

Mark Alber

Section Editor

PLOS Computational Biology

Reviewer's Responses to Questions

**Comments to the Authors:**

Reviewer #1: The comments are attached as a .pdf file.

Reviewer #2: Re: Review "Contributions of cell behaviour to geometric order in embryonic cartilage"

This article develops a three-dimensional model investigating the formation of columnar structure in early cartilage (and bone) formation during organism development. The model is motivated from a separate experimental / modelling paper (the model there was a cellular automaton on a lattice). In detail, the authors consider two scenarios: (1) the emergence of columns from dense collections of mesenchymal cells; and (2) the insertions of additional columns into an existing columnar structure (e.g.\\ tissue growth).

The authors goal is to elucidate the effect mechanical forces and cell division play in this process. For this purpose they propose a simple centre based model implemented in Chaste, with isotropic cell-medium friction, and cell-cell repulsion as the only cell-cell forces. Cells are placed in a three dimensional rectangle with a solid repulsive boundary placed below and above. The model setup resembles as cross-section of the modelled tissue configuration. Since previous modelling for this system was done using a lattice based model, mechanical insights were limited, which is being addressed in this article.

Through simulation studies the authors demonstrate that cell-cell repulsion with oriented cell division along the z-direction is sufficient for columnar structure to form, from an initially condensed group of cells (fairly flat). Further the authors demonstrate that oriented cell division is required, without columns do not form. The ''columnarness'' of their simulation outputs is quantified using an adequate order parameter. In addition, the authors investigate the effect and role other key parameters in their model play. Finally, the authors investigate whether their model supports the insertion of additional column. I suspect the motivation is to investigate what happens in a growing tissue? Their model cannot recapitulate the insertion of novel columns, hinting that additional mechanisms are required.

The paper is well written, all details are included, the code is available, and I feel confident that the results are reproducible. In my opinion this article provides a simple mechanical explanation for the formation of the observed columnar structure, the article clearly demonstrates limitations of their model, which highlight areas of future investigation. I do have some small remarks, and questions below, but I expect those to be easily addressable, and following revisions recommend this article for publication.

Comments

-----------

- I have three questions regarding the model for cell-cell division. As far as I understand cell division is implemented as follows. When the cell cycle time is completed two cells are placed a given distance apart along a given orientation.

Q1: Is the volume of the cells increased as they progress through the cell cycle?

Q2: Does simply placing the cells a distance r_0 apart immediately not lead to simulation challenges? If I were to implement this, I would let the cell double volume, then place two cells of the same volume at the same location, and then slowly push them apart to avoid numerical integration problems, and keep the system close to mechanical equilibrium. Is this not required since the cell population is always somewhat sparse?

Q3: In the simulations in which columns form, the division orientation is always along the z-axis i.e. fixed. Have the authors considered mechanical re-orientation of the cell division axis? In other words, let the mechanical interactions of the dividing cell determine the cell division axis? Or are the authors convinced that only the gradient of BMP guides the division axis?

- This comment is more of an observation. The authors demonstrate that oriented cell division is not sufficient to explain the intercalation of new columns into a pre-existing structure, unless sufficient space is available. This somewhat reminds me of classical Turing system results on growing domains see for instance Crampin et. al. (1999) i.e.\\ a new spike or stripe can be formed when sufficient space is available in the domain to accommodate it. Similarly, with the coarsening phenomena linked to growth (although there I do not completely understand the difference between sequential and continuous growth).

- Line 150: I follow the authors argumentation regarding why only a repulsive cell-cell interaction potential is used. But why is this interaction potential quadratic. Why not for instance, use a Hertz interaction potential or yet a different power? The authors finally explain their choice in the discussion. In my opinion it would make sense to explain their decision in the method or at least refer the reader to the discussion.

- I'm confused about with the authors description of the proliferation control. In detail, the paragraph beginning on line 184. Does this mean each cell can only divide a maximum of four times? If not how is the clonal envelope defined and computed?

- Clonal envelopes are referred to many times before they are defined. It's confusing.

- On line 218 the authors state that they do not impose any boundary conditions in the xy-plane. Does this mean that periodic boundary conditions are imposed?

- Lines in Figure 4d, 5d, 6d, 7d are not distinguishable in BW or for people with color seeing challenges. I would encourage the authors to employ different linestyles to avoid that challenge.

- Line 380: The authors hypothesize that column formation is aided by increased distance due to reduced mechanical repulsive forces. Would it be possible to quantify this interpretation?

- Line 382: Which is the perichondrial boundary again? Maybe just reiterate this as many readers will have forgotten at this point i.e.\\ that this is a discussion on the mechanical forces generated by planes located at z = u, l.

Reviewer #3: The review is uploaded as an attachment.

**Have the authors made all data and (if applicable) computational code underlying the findings in their manuscript fully available?**

Reviewer #1: Yes

Reviewer #2: None

Reviewer #3: Yes

PLOS authors have the option to publish the peer review history of their article (what does this mean?). If published, this will include your full peer review and any attached files.

Reviewer #1: No

Reviewer #2: No

Reviewer #3: No
---

## [Decision Letter · Decision Letter 1]

3 Nov 2023

Dear Miss Mathias,

We are pleased to inform you that your manuscript 'Contributions of cell behavior to geometric order in embryonic cartilage' has been provisionally accepted for publication in PLOS Computational Biology.

Best regards,

Philip K Maini

Academic Editor

PLOS Computational Biology

Mark Alber

Section Editor

PLOS Computational Biology

Reviewer's Responses to Questions

**Comments to the Authors:**

Reviewer #1: In my opinion the authors have made changes that were in line with suggestions of the reviewers, which in turn improved the quality of the manuscript.

Reviewer #2: The authors addressed my comments, and I have no further comments, and stand by my earlier recommendation that this article is publishable.

Reviewer #3: The authors have satisfactorily addressed all my comments, and the manuscript has greatly improved after revision.

**Have the authors made all data and (if applicable) computational code underlying the findings in their manuscript fully available?**

Reviewer #1: Yes

Reviewer #2: Yes

Reviewer #3: Yes

PLOS authors have the option to publish the peer review history of their article (what does this mean?). If published, this will include your full peer review and any attached files.

Reviewer #1: No

Reviewer #2: No

Reviewer #3: No

---

## [Editor Report · Acceptance letter]

23 Nov 2023

PCOMPBIOL-D-23-00028R1 

Contributions of cell behavior to geometric order in embryonic cartilage

Dear Dr Mathias,

I am pleased to inform you that your manuscript has been formally accepted for publication in PLOS Computational Biology. Your manuscript is now with our production department and you will be notified of the publication date in due course.

With kind regards,

Anita Estes
